# Monomer and Oligomer Transition of Zinc Phthalocyanine Is Key for Photobleaching in Photodynamic Therapy

**DOI:** 10.3390/molecules28124639

**Published:** 2023-06-08

**Authors:** Dafeng Liu, Longguang Jiang, Jincan Chen, Zhuo Chen, Cai Yuan, Donghai Lin, Mingdong Huang

**Affiliations:** 1MOE Key Laboratory of Spectrochemical Analysis & Instrumentation, Key Laboratory of Chemical Biology of Fujian Province, College of Chemistry and Chemical Engineering, Xiamen University, Xiamen 361005, China; 2College of Chemistry, Fuzhou University, Fuzhou 350002, China; 3State Key Laboratory of Structural Chemistry, Fujian Institute of Research on the Structure of Matter, Chinese Academy of Sciences, Fuzhou 350002, China

**Keywords:** photosensitizer (PS), photobleaching, photosensitizer aggregate, photosensitizer monomer

## Abstract

Photodynamic therapy (PDT) is recognized as a powerful method to inactivate cells. However, the photosensitizer (PS), a key component of PDT, has suffered from undesired photobleaching. Photobleaching reduces reactive oxygen species (ROS) yields, leading to the compromise of and even the loss of the photodynamic effect of the PS. Therefore, much effort has been devoted to minimizing photobleaching in order to ensure that there is no loss of photodynamic efficacy. Here, we report that a type of PS aggregate showed neither photobleaching nor photodynamic action. Upon direct contact with bacteria, the PS aggregate was found to fall apart into PS monomers and thus possessed photodynamic inactivation against bacteria. Interestingly, the disassembly of the bound PS aggregate in the presence of bacteria was intensified by illumination, generating more PS monomers and leading to an enhanced antibacterial photodynamic effect. This demonstrated that on a bacterial surface, the PS aggregate photo-inactivated bacteria via PS monomer during irradiation, where the photodynamic efficiency was retained without photobleaching. Further mechanistic studies showed that PS monomers disrupted bacterial membranes and affected the expression of genes related to cell wall synthesis, bacterial membrane integrity, and oxidative stress. The results obtained here are applicable to other types of PSs in PDT.

## 1. Introduction

Cancer remains a great threat to human health, and a strong research effort is underway to develop new approaches to resolve this issue [1,2,3]. Photodynamic therapy (PDT) is an established therapeutical modality for the treatment of various cancers (e.g., prostate, breast, lung, head and neck, skin, and pancreas) [4], and is approved by many government agencies, including the US Federal Drug and Food administration and others. In PDT, a photosensitizer (PS) absorbs light with specific wavelengths and ultimately generates reactive oxygen species (ROS), such as singlet oxygen (^1^O_2_) or hydroxyl radical (·OH), creating cytotoxicity to rapidly kill nearby cancer cells [5,6]. PDT is endowed with several favorable features for the treatment of cancer, such as non-invasiveness, the absence of complications arising from surgery, the minimization of side effects characteristic of chemotherapy and radiotherapy, and the lack of need for long-term inpatient treatment [4,5,6].

ROS also degrade the photosensitizer itself (Figure 1a), a process called ‘photobleaching’, leading to a reduction or even loss of photodynamic efficacy. Photobleaching is a complicated process and is affected by many parameters, such as light (irradiation dose and wavelength), the photosensitizer itself (chemical structure and concentration), and oxygen (oxygen concentration). For example, Georgakoudi and Foster [7] found that the photobleaching of the photosensitizer PpIX started from the regions closest to the oxygen source. They studied PpIX photobleaching in a tumor spheroid, demonstrating the importance of oxygen. Dysart et al. [8] observed that higher photosensitizer concentrations led to faster photobleaching of the photosensitizer mTHPC. In another interesting example, the rate of photobleaching was found to increase as irradiance dose was reduced during the study of a multicell spheroid photosensitized by mTHPC [9]. Similarly, many authors found that the photobleaching of photosensitizers was rapidly increased under light fractionation where continuous irradiation was punctuated by dark intervals to allow the re-oxygenation of anoxic regions [10,11], showing that light and oxygen synergistically affect photobleaching. Overall, the photobleaching of photosensitizers is a complicated process in PDT. Much effort has been devoted to minimizing photobleaching by retaining photodynamic efficiency of photosensitizers during PDT.

In this work, we study the balance of photodynamic and photobleaching effects during PDT using a zinc phthalocyanine type PS (ZnPc). We used a microorganism as a research subject for the ease of the study. The rapid and worldwide appearance of multidrug-resistant microbes has become a major global threat to human health, and PDT is strongly advocated as an efficient and effective antibacterial therapy to overcome such threats [12]. Our results demonstrate that the PS formed aggregates in an aqueous solution, which did not inactivate the photodynamic activity but was resistant to photobleaching. Upon contacting the bacterial surface, the PS aggregate was partially disintegrated, releasing the PS monomer which was photodynamically inactive and subsequently photobleached. Interestingly, we also found that light illumination also facilitated the disassembly of aggregated PS to release the PS monomer, leading to enhanced photodynamic effects of PDT. The PS aggregation had a lack of photodynamic action towards bacteria via an intermediate-PS monomer under illumination. We further studied the antibacterial mechanism of the PS aggregate and found that the PS monomer disrupted the integrity of the bacterial membrane and induced a profile of gene expression. Therefore, our results demonstrate that the PS aggregate notably enhanced photo-stability and bound the PS aggregate to bacteria acting as a store for the PS monomer, which was generated when the PS aggregate encountered bacteria and/or light. Such conclusions are not only valid for bacterial surfaces, but most likely for cancer cell surfaces as well.

## 2. Results

### 2.1. Photosensitizer Aggregate Possessed Neither Photobleaching nor Photodynamic Effect

In this study, we used a photosensitizer (PS), ZnPc(Lys)_5_ (Figure 1b, Appendix A), with two key features: a zinc phthalocyanine (ZnPc) group to provide a potent PDT effect in generating ROS and, to overcome the aqueous solubility issue of ZnPc, a pentalysine moiety [13,14,15]. In an aqueous solution, this PS formed aggregates with hydrophobic ZnPc in the core and exposed hydrophilic pentalysine in the shell (Figure 1b) [16].

This aggregated PS showed no measurable fluorescence (Figure 2a) but showed a strong adsorption intensity at 630 nm (Figure 2b). In addition, the aggregated PS showed no photodynamic effect [16]. The aggregated PS, however, was quite photostable and no photobleaching at all was observed at increasing light doses up to 3.13 J/cm^2^, due to the fact that it had no change in UV/Vis absorption (Figure 2c) and did not fluoresce (Figure 2d).

### 2.2. Monomeric Photosensitizer Is Photodynamically Active but Can Be Photobleached

The PS aggregate in an aqueous solution was disassembled into a PS monomer with either detergent (0.05% tween20) or a lysis buffer (0.1 M NaOH/1% SDS). The PS monomer showed a main absorption at 678 nm (Figure 3a) and strong fluorescence (l_exc_ = 610 nm, l_em_ = 680 nm) (Figure 3b) [13,14]. This PS monomer was significantly photobleached (Figure 3c,d) under conditions identical to that used for the PS aggregate (Figure 2). The photobleaching rate of the PS monomer was up to 41.84% at a light dose of 3.13 J/cm^2^ (Figure 3e). Nevertheless, the PS monomer rather than the PS aggregate showed photodynamic efficacy (see below).

### 2.3. Aggregated PS in Bacterial Suspension Showed Highly Photodynamic Efficacy

We studied if the PS aggregate possessed antibacterial activities by determining the minimum inhibitory concentration (MIC, the concentration resulting in no growth of bacteria) and the minimum bactericidal concentration (MBC, the concentration at which no more than five colonies were observed on the plates). For MIC measurements, the PS aggregate at increasing concentrations was added to bacterial suspensions, incubated, and illuminated (Appendix A), and MICs were measured based on bacterial growth. For MBC measurement, a 100 μL mixture of the PS aggregate and bacteria was transferred to culture plates for colony counting after 37 °C incubation for 24 h. We found that the aggregated PS killed bacteria effectively; the MICs of the PS aggregate against *E. coli* and *S. aureus* were 1.6 and 0.8 µM, respectively, and the corresponding MBCs were 3.2 and 1.6 µM in bacterial suspension (Table 1).

Since the aggregated PS did not have a PDT effect as shown in the first Results, we hypothesized that the PS aggregate inactivated bacteria in the solution by releasing the PS monomer, which then generated ROS to kill nearby bacterial cells. Therefore, we carried out the following mechanistic studies to examine this hypothesis.

### 2.4. Membrane Surface Triggered the Disassembly of PS Aggregate to Release PS Monomer

We studied the aggregation state of the PS on a bacterial surface by measuring the fluorescence emission spectra and UV/Vis absorption of the PS aggregate bound on the bacterial surface. We found that the 678 nm peak of the bound PS on the bacterial surface became more prominent (Figure 4a,b) compared to the solution phase (Figure 2c), showing the presence of the PS monomer bound to bacteria. In addition, the PS aggregate become fluorescent upon binding to bacteria (fluorescence intensity of 723 a.u. for *E. coli* and 1564 a.u. for *S. aureus*), which was in good agreement with our previous measurement [13,16]. These results indicated that bacteria caused the disassembly of the PS aggregate on the bacterial surface to generate the PS monomer, which shows both photobleaching (Figure 3c,d, Appendix A) and photodynamic activity against bacterial cells (Table 1).

### 2.5. LED Light Facilitated the Disassembly of PS Aggregation and the Generation of More PS Monomers

In these sets of experiments, the PS aggregate was incubated with bacteria and the unbound PS was washed out, followed by light illumination (0.34 J/cm^2^). We found that the 678 nm peak of the bacteria-bound PS aggregate became stronger as compared with the no-illumination control group (Appendix A). This demonstrated that light facilitated the generation of the PS monomer on the bacterial surface. This was a surprising finding considering that phthalocyanine, as a member of organic dyes, tends to be bleached in the presence of light.

These results were further validated by the fluorescence measurements of the PS aggregate bound on bacteria after illumination with different light doses (0 to 1.36 J/cm^2^, Figure 4). The fluorescence signal was proportional to the amount of the PS monomer because only the PS monomer but not the PS aggregate fluoresces [13] (Figure 2a and Figure 3b). In parallel experiments, we also determined the total amount of bound PS by lysing the PS aggregate-bound bacteria. We found that the fluorescence intensities of the PS increased with the increase of light doses (Figure 4c,f). The total amount of bound PS (Figure 4d) was much more than that of the PS monomer (Figure 4c, fluorescence intensities 1000–2000 vs. 600–900 for *E. coli* based on fluorescence intensities). The total amount of PS bound on the cell surface decreased as the light dose increased (Figure 4d). Similar trends were obtained in *S. aureus* (Figure 4f,g). Thus, we concluded that light facilitated the generation of the PS monomer on the bacterial surface from the bound PS aggregate. We further calculated that fractions of the PS monomer were from 32.6 to 73.2% for *E. coli* (Figure 4e) and 45.0 to 87.6% for *S. aureus* (Figure 4h) at increasing light doses. More PS monomers on the bacterial surface led to the enhancement of the photodynamic effect toward the bacteria under illumination. We wanted to point out that the PS monomer itself was eventually photo-destroyed by generated ROS, resulting in the decline of the total bound PS aggregate upon increasing light doses (Figure 4d,g).

### 2.6. Mechanistic Studies–PS Aggregate Reached Fast Saturated Binding to Membrane Surface in Only 3–6 min

We developed a flow cytometry method to monitor the binding kinetic process of the PS aggregate by fluorescence intensity of the PS monomers, which were bound to bacteria and generated from the PS aggregate [17]. In addition, we prepared bacteria without cell walls (named spheroplasts) using lysozyme from intact bacteria (Figure 5c) for the purpose of further mechanistic studies. Spheroplasts retain bacteria plasma membranes with the bacterial cell wall removed [18,19,20]. Spheroplasts are widely used in mechanistic studies, including patch-clamp and fusion [21]. We mixed intact bacteria or spheroplasts with the PS aggregate (3 µM) (Appendix A) and monitored binding kinetics every 30 s using the fluorescence signal of the PS monomers. We found that the amount of absorbed PS aggregate increased with incubation time (Figure 5a,b). The binding of the PS aggregate to intact *E. coli* reached a plateau at 2.5 min, slightly faster than the corresponding spheroplasts (3 min, Figure 5a). For *S. aureus*, the binding rates were slightly slower than *E. coli* (4.5 min for intact *S. aureus* and 6 min for spheroplasts, Figure 5b). These fast adsorption times (sec to min) were in line with previously reported values [13,17]. The results also showed that the adsorption rates of intact cells were faster than the spheroplasts. This is likely due to the fact that bacterial cell walls have more negative charges compared to membranes [22,23]. In the following experiments, we used an incubation time of 3 min for *E. coli* and 6 min for *S. aureus* in order for the PS aggregate to be fully adsorbed onto the intact bacteria or spheroplasts.

### 2.7. Mechanistic Studies–Partition of PS Aggregate between Bacterial Plasma Membrane and Bacterial Wall

We first measured the binding affinities of the PS aggregate on bacteria. The PS aggregate at increasing concentrations (0.039 to 40 µM) was added to bacterial suspensions and incubated with bacteria, and the total bound PS aggregate was quantified after washing out the unbound PS aggregate and lysing bacteria. The total bound PS molecules increased (0.1 × 10^5^ to 7.9 × 10^5^ per *E. coli* cell, and 0.1 × 10^5^ to 9.0 × 10^5^ per *S. aureus* cell) upon increasing PS aggregate concentrations (Figure 6a,b). The PS aggregate binding to *E. coli* was slight tighter than to *S. aureus* (equilibrium constant (K_d_)—2.5 µM for *E. coli* vs 1.0 µM for *S. aureus*, Figure 6a,b). The number of PS aggregates bound to bacteria was the opposite: more on *S. aureus* than on *E. coli* (maximum adsorption (B_max_)—9.1 × 10^5^ vs. 8.2 × 10^5^, Figure 6a,b).

The bacteria used bacterial walls and plasma membranes as the barrier. To identify the preferred target of the PS, the PS aggregate at increasing concentrations (0.2 to 25.6 µM) was added to either the spheroplast or intact bacteria suspensions. The bound PS was measured after the solution PS was washed off. Now the total PS molecules were found to be 0.2 × 10^5^ to 3.7 × 10^5^ molecules per spheroplast of *E. coli* upon increasing the PS aggregate concentrations (Figure 6c), and there were 0.3 × 10^5^ to 4.8 × 10^5^ molecules per spheroplast of *S. aureus* (Figure 6d). In addition, we found that the total number of PS molecules bound to the bacterium of *S. aureus* (0.6 × 10^5^ to 9.1 × 10^5^) was higher compared to *E. coli* (0.4 × 10^5^ to 7.9 × 10^5^) as the PS aggregate concentrations increased (Figure 6c,d). These results also indicate that the number of PS molecules absorbed on spheroplasts was about half of the PS molecules bound to the intact bacterium. Another half was presumably bound to the bacterial lipid bilayer membrane. Our result was in good agreement with the previous measurement (~10^5^ molecules per bacterium at 1 µM PS) [13,24].

### 2.8. Mechanistic Studies–Cell Membrane Integrity

The membrane integrity was evaluated by determining the fluorescence of anilinonaphthalene-8-sulfonic acid (ANS, molecular mass of 299.34 Da) in the presence of the PS aggregate at different concentrations. ANS penetrates into the phospholipid bilayer of the bacterial membrane when the membrane is disrupted, resulting in the dramatic increase of its fluorescence intensity. In our experiments, the PS aggregate at increasing concentrations was incubated with *E. coli* for 3 min, followed by illumination (12.7 J/cm^2^). ANS (0.565 mM) was then added to the bacterial suspensions and its fluorescence spectra were recorded. This procedure avoided the degradation of the probe by photo-generated ROS. We found that ANS showed a blue shift of maximum emission and enhancement of fluorescence intensity with the increase of PS aggregate concentrations in the presence of light (Figure 7a). Emission maximums increased (476, 486 to 502 nm) with PS aggregate concentrations, and the fluorescence intensities of the emission peak also increased (2475, 3585 and 4905 a.u.) (Figure 7a), both indicating that the probe ANS relocated into a relatively hydrophobic environment (bacterial membrane), presumably due to the membrane disruption by photodynamic effect. The bound PS aggregate was further disassembled into smaller PS monomers by contacting the membrane and/or by illumination (Figure 7a), demonstrating that the photodynamic effects increased with more PS.

As a control, the emission maximum of the probe was found to be not shifted, and its fluorescence intensity did not change in the dark, demonstrating that ANS did not penetrate the phospholipid bilayer, showing no dark toxicity of the PS aggregate; this was in line with the results of growth curves of bacteria treated with PS aggregate in the dark (Appendix A).

We also measured the membrane permeability of intact *E. coli* and *S. aureus* cells using another dye N-phenyl-1-naphthylamine (NPN, molecular mass of 219.29 Da). NPN can penetrate the damaged cytoplasmic membrane of bacterial cells and shows strong fluorescence (ex = 350 nm, em = 420 nm). Bacterial suspension was incubated with a PS aggregate and was illuminated by light, and then the probe NPN was added to the suspension in the absence of light. The fluorescence intensity of the probe was then measured. We found that bacterial membrane permeability was increased in *E. coli* and *S. aureus* under illumination (Figure 7b,c), demonstrating that the photodynamic activity of PDT was increased with the increase of PS monomer concentrations. In the presence of light, the addition of the low-dose PS aggregate (below 1/5 × MIC) to *E. coli* suspensions initially led to a dramatic increase of the fluorescence intensity of the probe (Figure 7b). The fluorescence intensity reached a maximum emission at 3/5 × MIC, and then decreased for *S. aureus* (Figure 7c). For *E. coli*, the membrane permeability levels were significantly higher compared to *S. aureus*. On the other hand, the dose response trend was almost the same, except the permeability maximized at 1/5 × MIC concentration.

### 2.9. Mechanistic Studies–PS Aggregate Released Its Monomer to Induce Altered Expression Profiles of Related Genes

In order to investigate the antibacterial mechanism of the PS aggregate interaction with bacterial cell walls and membranes at the molecular level, the expression of different genes was measured using RT-qPCR at various PS aggregate concentrations. *S. aureus* was used to measure the expression of genes related to the cell wall and plasma membrane, owing to the prominent structural features of the bacterial cell wall in Gram-positive bacteria (thicker layers of peptidoglycan compared to Gram-negative bacteria). In order to measure qPCR signals without killing all *S. aureus*, we used two strategies: (1) Using the PS aggregate at a low concentration (1/2-fold MIC of PS aggregate against *S. aureus*); (2) employing low light doses (0, 2, 4 and 6 J/cm^2^).

Two target genes, srtA and murZ, which encode proteins that are involved in cell wall biosynthesis, were chosen. The SrtA gene is responsible for cross-linking surface proteins to peptidoglycan [25,26]. The gene murZ encodes transferase enzymes, facilitating the synthesis of enolpyruvyl UDP-N-acetylglucosamine (EP-UDP-GlcNAc) [27,28]. We found that both genes were up-regulated by PS monomers with the increase of light doses (Figure 7d), reflecting enhanced bacterial cell wall biosynthesis during PDT.

Bacteria plasma membranes may be damaged by PS monomers. The expression of the mprF gene was measured. MprF encodes an enzyme that is responsible for the addition of lysine to phosphatidyl glycerol, forming the positively charged species lysylphosphatidyl glycerol [29,30,31]. We found that the mprF gene increased to a larger extent with the increase of light doses (Figure 7d), indicating that the plasma membrane was damaged by the PS monomer with photodynamic effects, which was in good agreement with hydrophobicity and the zeta potential of the bacterial surface (Appendix A).

On the other hand, we used *E. coli* to measure the expression of genes involved in oxidative stress. In order to induce gene expression and avoid overkilling, we used (1) three sessions of illumination separated by 2 h and (2) a low concentration of the PS aggregate (1/8-, 1/4- and 1/2-fold of MIC value). SodA is a key structural gene in bacteria, and it expresses superoxide dismutase under redox stress to protect bacterial cells [32,33,34]. OxyR is a transcriptional regulator, and it is capable of enhancing anti-oxidative defenses by inducing the expression of catalases and peroxidases [33,35]. While the recA gene plays a major role in DNA repair, it also reduces the expression levels of lexA [36]. We found that the transcription levels of oxidative stress-related genes were up-regulated at increasing PS aggregate concentrations (Figure 7e). Nevertheless, lexA expression was down-regulated (Figure 7e). These differential expression profiles of these genes indicated that gene repair was initiated, and that monomeric PS indeed damaged DNA. In addition, espA expression also was measured. EspA is a virulence gene in bacteria [33,37]. The secreted protein EspA plays a key role in the process of host cells being infected by bacteria. We found that espA was down-regulated upon increasing PS aggregate concentrations (Figure 7e), showing that bacterial cell viability reduced under monomeric PS.

Put together, these results demonstrate that PS monomers generated from the disassembly of a PS aggregate (Figure 8a,d) evoked ROS (Figure 8e) to photo-dynamically damage bacteria barrier functions and to overload the bacteria anti-oxidative and DNA repair capabilities, leading to bacteria death.

## 3. Discussion

In this work, we reported a type of PS aggregate possessing neither photobleaching nor photodynamic effects, measured its antibacterial activities, and quantified the total amount of bound PS to bacteria. Interestingly, we found that the bacteria induced the disassembly of the PS aggregate into PS monomers (Figure 8a–c), and light facilitated the disassembly into more PS monomers on the bacterial surface (Figure 8d), which generated ROS under illumination to kill bacteria (Figure 8e). We also studied the photodynamic mechanism of the PS aggregate and found that the integrity of the bacterial membrane was disrupted and a profile of related gene expression was induced. It appears that the PS aggregate served as a reservoir for the slow release of PS monomers in PDT.

As organic dyes, PSs suffer from limited photostability, leading to undesired photobleaching and then the compromised antibacterial effect of PDT. Therefore, it is important to balance the two parameters between photobleaching and photodynamic effects. Many efforts have been devoted to designing and preparing an ideal PS, which would have no photobleaching in order to possess maximal photodynamic efficacy in PDT. The photobleaching of PSs depends upon a large number of synergistic parameters, such as light delivery and dosimetry, fluence rate, drug-light interval, light fractionation, light wavelength, PS concentration, PS delivery, oxygen concentration, and so on. On the other hand, the photobleaching of PSs can proceed via ·OH- and H_2_O_2_-mediated Type-I reactions or ^1^O_2_–mediated Type-II reactions (or both), with one or the other predominating according to different PSs [38]. However, the relative contribution of the Type-I and -II reactions in the solution phase is also strongly dependent on the experimental conditions, such as the solvent used or the presence of biological substrates, including proteins [39,40]. Therefore, it is difficult to consider all the parameters to avoid the photobleaching of the PS without compromising the photodynamic effects.

Some strategies in retaining PS photostability are reducing the number of reactive species in the environment of PDT (e.g., under low oxygen concentrations), reducing the reactivity of target sites of PS together with their intermolecular and intramolecular protection, and so on. For example, a dramatic increase of photostability was achieved by incorporating substitutions in the central part of symmetric near-IR carbocyanine dye. The photostability of 3-hydroxyflavones can be improved by substituting central oxygen into nitrogen to make 3-hydroxyquinolones [41]. A new family of fluorophores with photostability were created by the replacement of oxygen with a nitrogen atom in the dicyanomethylene-4H-pyran (DCM) moiety [42]. In our work, the aggregation of the PS was used to evade its photobleaching. However, PS aggregates themselves show neither a photodynamic effect nor photobleaching, but they retain the PDT effects of PSs by releasing PS monomers, which possess photodynamic effects against bacterial cells. One of the challenges faced by photodynamic therapy is the photo-degradation of PSs themselves. ROS are generated in the presence of PS monomers, which creates cytotoxicity, killing the pathogen in the illuminated area. At the same time, ROS degrade the PS itself because most of the PSs are organic dyes themselves [43,44,45]. High light fluency (100–200 J/cm^2^) was typically used in photodynamic cancer therapy in clinics. Organic dyes are vulnerable under intensive light or high light fluency. For example, the antibody-conjugated fluorescein had a half-life of only 20 s under a typical confocal microscope [46].

On the other hand, the details of the disassembly of PS aggregates require further study, and comprise an important factor to consider for the design of the molecular structure of newer generations of PSs. Such PSs typically have a prominent chemical structural feature of large aromatic rings, which are hydrophobic in nature. Thus, many PSs tend to form some degree of aggregate by themselves in an aqueous solution, presumably through π-π stacking, leading to reduced photodynamic efficacy. Detergents are typically used to solubilize the PS during clinical application to increase PS solubility. For example, castor oil is one type of detergent typically used. However, the detergent caused patients pain during PDT treatment. We show here that PS aggregation is another way to solubilize PS without compromising its PDT efficacy. Growing cancer cells have fast and distinguished metabolisms (Warburg effect) and present negative charges on their cell surfaces that are higher than those of normal cells [47], although they may not be as high as those of bacterial surfaces. Such cell surface membranes may promote the dissociation of PS aggregation into monomeric PSs in a way similar to what we report here.

The finding in this work is likely to have applicability to dye discoloration treatment in the textile industry. More than 10% of total textile dyes produced or about 700,000 tons of dye per year [48,49] are released. Some synthetic dyes may be carcinogenic or mutagenic. Biotechnological discoloration processes using fungi are one way to reduce the dye toxicity by their non-specific, lignin-degrading enzymes which degrade a wide range of organic pollutants, including textile dyes [49].

## 4. Materials and Methods

### 4.1. Light Source

The planar light-emitting diode (LED) light source contains 24 lamps emitting 660 nm red light with a bandwidth of 25 nm and a fluence rate of 41.67 mW/cm^2^ [17,50,51]. With 1 min of illumination, this LED generates a light dose of 2.5 J/cm^2^.

### 4.2. Bacterial Cell Preparation

Gram-negative bacteria *Escherichia coli* (*E. coli*, ATCC 25922) and Gram-positive bacteria *Staphylococcus aureus* (*S. aureus*, ATCC 6358) were used as model organisms for antibacterial experiments. The Gram-negative bacteria, *E. coli DH5α*, was transformed with bioluminescent plasmid (pAKlux2.1) in our laboratory [17]. The bacterial strains were grown in a Luria-Bertani (LB) culture medium at 37 °C.

### 4.3. Spheroplast Preparation

Spheroplasts remain bacteria plasma membranes after bacterial cell walls are removed [18,19,20]. Spheroplasts were widely applied to patch-clamp, fusion, and other experiments and also antibiotic studies [18,21,52,53,54,55].

Spheroplasts of *E. coli* were prepared using a previously reported method [18,19,20,21,53,54] with some modification. The bacterial cells were allowed to grow in an LB medium at 37 °C and then were harvested, washed, and suspended in 0.1 M Tris-HCl (pH 7.2), 0.3 M sucrose, and 1 mM ethylenediaminetetraacetic acid (EDTA). Bacterial suspension stood for 15 min at room temperature. Lysozyme (600 µg/mL) was added to the bacterial suspension, which was then incubated at 37 °C for 45 min and 150 rpm. The spheroplasts were collected by centrifugation and then re-suspended in a spheroplast buffer (0.1 M Tris-HCl, 0.3 M sucrose and 0.1 M KCl (pH 7.2)).

For *S. aureus*, spheroplasts were prepared using a procedure similar to that of the spheroplasts of *E. coli*. The bacterial cells were allowed to grow in an LB medium at 37 °C and then were harvested, washed, and suspended in 0.1 M Tris-HCl (pH 7.2) and 0.3 M sucrose. Lysozyme (800 µg/mL) was added to the bacterial suspension. After 60 min of incubation at 37 °C and 150 rpm, spheroplasts were obtained by centrifugation and then re-suspended in a spheroplast buffer.

### 4.4. Synthesis of ZnPc(Lys)_5_

Monosubstituted β-carboxyphthalocyanine zinc (ZnPc-COOH) was first synthesized according to our published protocol [13,15,24,56]. A polylysine ZnPc conjugate of five charges (ZnPc(Lys)_5_) was then prepared by coupling ZnPc-COOH prepared as above with tBoc-protected polylysine on Wang resin ((Lys)_5_-Wang resin; Shanghai Biotech Bioscience and Technology Co. Ltd., Shanghai, China). Typically, 0.5 g ZnPc-COOH (800 µmol) was dissolved in N,N-dimethylformamide (DMF, 25 mL) with 5% pyridine and was incubated in the dark with the peptide coupling agent N,N’-dicyclohexylcarbodiimide (DCC, 330 mg, 1.6 mmol) and the activating reagent 1-hydroxybenzotriazole (HOBt, 215 mg, 1.59 mmol) for 30 min at room temperature. This solution was then mixed with filtered (Lys)_5_-Wang resin solution which was prepared as follows: 375 mg of (Lys)_5_-Wang resin (parity grade: 0.3–0.8 mmol/g) was added into DMF (6 mL) and the solution was kept at room temperature for 30 min before being filtered. After adding another 4 mL of DMF with 5% pyridine, the mixed solution was kept stirring overnight to allow the reaction to reach completion. The Wang resin was washed with DMF until becoming colorless and then washed with dichloromethane and methanol before being dried under a vacuum. To cleave the coupled product from the resin and to cleave tBoc-protecting groups from lysine residues, the resin was allowed to react with 95% trifluoroacetic acid for 3 h. ZnPc(Lys)_5_ was then precipitated out by cooled anhydrous ethyl ether. These final products were dried under a vacuum and further purified by preparative HPLC (Dalian Elite Analytical Instruments Co. Ltd., Dalian, China) with a Fuji C_18_ column (250 × 30 mm, 10 μm, Beijing Chuangxin Tongheng Science & Technology Co., Ltd., Beijing, China), eluting with a linear gradient of 50→100% mixed solvent CH_3_OH/CH_3_CN (1:1 *v*/*v*) under a flow rate 10 mL/min within 20 min. The final yield of ZnPc(Lys)_5_ was 38.6% based on the compound ZnPc-COOH.

### 4.5. Residual PS Aggregation Measurements

PS aggregation at 1 µM was added to a bacterial suspension with PBS or 0.05% Tween20 and was incubated with bacteria. The total PS aggregation bound to bacteria was irradiated with different light doses (0, 2, 4, 6, 8, 10, 12, 14, 16, 18, and 20 J/cm^2^) after the unbound PS aggregation was washed away. The PS aggregation bound to bacteria was in a lysis buffer for 60 min. Fluorescence emission spectra (λ 640–750 nm) were measured with step 1 nm using Synergy4^TM^ (BioTek, Shoreline, WA, USA) [13]. Residual rates and first-order rate constant [57,58,59,60] of PS aggregation resulted from these data of fluorescence intensities at l_em_ = 680 nm with various light doses [59,60]. The excitation wavelength of the PS aggregation was set at 610 nm. The residual rate (%) was calculated by the equation:Residual rate = C_t_/C_0_ × 100%,
where C_t_ is the fluorescence intensity for different time of illumination and C_0_ is the fluorescence intensity without illumination.

In addition,
k = −1/t × ln(C_t_/C_0_)
where k is the apparent first-order rate constant and t is illumination time.

### 4.6. Bacterial Uptake of PS Aggregation by Bacteria

We measured bacterial uptake of PS and PS aggregation using a previously reported method [13,50] with some modifications. PS at 1 µM was added to bacterial suspensions (3.6 × 10^8^ CFU/mL) containing 0.05% tween20. On the other hand, a PS aggregation at 1 µM was added to bacterial suspensions (3.6 × 10^8^ CFU/mL) in PBS. The PS aggregation in the bacterial suspension was then washed away with lysis with 0.1 M NaOH/1% sodium dodecyl sulfate (SDS) to give a homogeneous solution. The fluorescence of the cell extract was measured on a microplate reader (Synergy 4, BioTek Instruments) at l_exc_ = 610 nm and l_em_ = 680 nm. Standard curves were made with known added amounts of PS dissolved in lysis buffer. Results were expressed as the number of PS aggregations per bacterium.

### 4.7. Photostability Measurements

PS aggregation suspensions were irradiated with different light doses (0, 0.63, 1.25, 1.88, 2.5, and 3.13 J/cm^2^) after the PS aggregation at 1 µM was added to a 0.05% tween20 buffer. UV/Vis absorption (λ 500–800 nm) and fluorescence emission spectra (λ 640–750 nm) were recorded with step 1 nm using Synergy4^TM^ (BioTek, USA) [13]. Photobleaching rates of PS aggregation resulted from these data of fluorescence intensities at l_em_ = 680 nm with various light doses. The excitation wavelength of the PS aggregation was set at 610 nm. For photostable measurement of the PS aggregation in phosphate-buffered saline (PBS), the method was the same as above. The photobleaching rate (%) was calculated by the equation:Photobleaching rate = 1 − A/B × 100%, 
where A was the fluorescence intensity with illumination and B was the fluorescence intensity without illumination.

### 4.8. Binding Kinetics of PS Aggregation onto Bacteria

The binding kinetics were monitored using a flow cytometer (BECKMANCOULTER) with a fluorescent channel (APC-A750-A, ex = 638 nm, em = 780 ± 60 nm) [13,17]. Intact bacteria or a spheroplasts culture were centrifuged and re-suspended by 50-fold dilution. The instrument gain and threshold population (<2%) were found out using the FSC-SSC plot. The SSC-H and APC-A750-A plots were also recorded to find out the gating value for the basal fluorescence signal. To monitor the kinetics of the PS aggregation binding to intact bacteria or spheroplasts, the PS aggregation was added to an eppendorf tube containing intact bacteria or spheroplasts to make a 3 µM PS aggregation concentration, and the fluorescence signals were acquired right away every 30 s. Such measurements were carried out three times for each PS aggregation and the results were averaged.

### 4.9. Antibacterial Activity Test

Minimum inhibitory concentrations (MICs) and minimum bactericidal concentrations (MBCs) were determined by the double dilution method [61,62,63,64,65]. Logarithmic bacterial strains were used in this experiment. Inocula of bacterial strains were prepared by adjusting overnight cultures to contain 1.73 × 10^7^ CFU/mL in an LB medium. The PS aggregation was incubated with bacteria (pH7.2), and the incubation time was as follows: (1) Binding plateau time. The incubation time of *E. coli* with the PS aggregation was 3 min, but 6 min for *S. aureus*. (2) The incubation times of *E. coli* and *S. aureus* with the PS aggregation were the same: 30 min. Aliquots of 100 μL of inocula (10^6^ CFU/mL) were mixed with a PS aggregation of 900 μL of serial twofold dilutions in centrifuge tubes and then incubated with shaking at 37 °C for 20 h in the dark after a photoirradiation dosage of 12.7 J/cm^2^. The MIC was defined as the lowest concentration of antibacterial materials where no growth was observed. Then 100 μL from each of the clear centrifuge tubes was transferred to culture plates to incubate for 24 h at 37 °C in the dark. The colonies were calculated and compared to control plates. The survival rate (%) was calculated as follows: counts of samples treated with PS aggregation/counts of control. The plate containing the lowest concentration of PS aggregation showing no growth indicated minimum bactericidal concentration (MBC). All the experiments were carried out in quintuplicate.

### 4.10. Fractions of PS Aggregation on Bacterial Surface

To explore the fraction of PS, UV/Vis absorption spectrum and fluorescence intensity of the PS aggregation were measured with step 1 nm using Synergy4^TM^ (BioTek, USA) [13,14,24,50]. The PS aggregation at 1 µM was added to bacterial suspensions (10^7^ CFU/mL) and then was incubated. The unbound PS aggregation was washed away in order for only the PS aggregation to remain bound to the bacteria. The UV/Vis absorption spectrum of the PS aggregation in the PBS or spheroplast buffer was recorded from 500 to 800 nm. On the other hand, the fluorescence intensity (l_exc_ = 610 nm, l_em_ = 680 nm) was calculated after the bacteria absorbing the PS aggregation were dissolved in 0.1 M NaOH/1% SDS for 1 h.

### 4.11. Fraction of Bound PS Aggregation under Illumination

The PS aggregation at 1 µM was added to bacterial suspensions and incubated with bacteria. The unbound PS aggregation was washed away, and then illumination (0 and 0.34 J/cm^2^) was carried out. The UV/Vis absorption spectra were recorded from 500 to 800 nm.

The PS aggregation at 5 µM was added to 1 mL bacterial suspensions (10^7^ CFU/mL) and incubated with bacteria. The unbound PS aggregation was washed away using a PBS buffer. Fluorescence emission spectra (660 to 710 nm) of the bound PS aggregation in the PBS buffer were measured at l_exc_ = 610 nm after the PS aggregation that was adhered to bacteria was treated with various light doses (0, 0.34, 0.68, and 1.36 J/cm^2^). On the other hand, illumination was conducted with different light doses (0, 0.34, 0.68, and 1.36 J/cm^2^) before the PS aggregation absorbed to bacteria was in 1 mL 0.2 M NaOH/2% SDS for 1 h. The fluorescence spectra (660 to 710 nm) were recorded with l_exc_ = 610 nm.

### 4.12. Measurement of Spheroplasts Yield

We measured the yield of *E. coli* or *S. aureus* spheroplasts at different times (0, 2, 4, 6, 8, and 10 min) after the spheroplasts were prepared in 0.1 M Tris-HCl, 0.3 M sucrose, and 0.1 M KCl, pH 7.2 (spheroplast buffer). The spheroplasts of *E. coli* or *S. aureus* were re-suspended with double-distilled water (ddH_2_O) for 20 min after the spheroplasts were prepared. A dilution was respectively spread onto culture plates to grow overnight under an anaerophilic environment at 37 °C. The colonies were calculated and compared to control plates to determine the yield of spheroplasts. Measurements of spheroplast yields were carried out in quintuplicate. Spheroplast yield (%) was calculated by the equation:Spheroplasts yield = 1 − Y × 100%,
where Y is the survival rate of bacteria.

### 4.13. The Optimal Time for Adding PS Aggregation to Spheroplasts Suspension

We measured the optimal time when the PS aggregation was added to the spheroplast suspension after the spheroplasts of *E. coli* or *S. aureus* were prepared. The PS aggregation (1 µM) was added to the spheroplast suspension (10^7^ CFU/mL) at different standing times (0, 2, 4, 6, 8, and 10 min). The PS aggregation in spheroplasts or intact bacteria suspension was washed away after the PS aggregation was incubated with spheroplasts or intact bacteria at 37 °C. The PS aggregation absorbed to spheroplasts or intact bacteria was dissolved in 0.1 M NaOH/1% sodium dodecyl sulfate (SDS) for 60 min to give a homogeneous solution. The fluorescence of the cell extract was measured on a microplate reader (Synergy 4, BioTek Instruments) at l_exc_ = 610 nm and l_em_ = 680 nm. Measurements of the time determinations were carried out in quintuplicate. Results were expressed as a fluorescence ratio. The fluorescence ratio was calculated by the equation:Fluorescence ratio = M/N × 100%,
where M is the fluorescence intensity of the PS aggregation bound to the spheroplasts and N is the fluorescence intensity of the PS aggregation bound to intact bacteria.

### 4.14. Bacterial Uptake of PS Aggregation by Bacteria

We measured the bacterial uptake of the PS aggregation using a previously reported method [13,50] with some modifications. Different PS aggregation concentrations (0.039 to 40 µM) were added to bacterial suspensions (3.56 × 10^8^ CFU/mL). The incubation of the PS aggregation with bacteria was conducted at pH7.2. The PS aggregations in bacterial suspension were then washed away before lysis with 0.1 M NaOH/1% sodium dodecyl sulfate (SDS) to give a homogeneous solution. The fluorescence of the cell extract was measured on a microplate reader (Synergy 4, BioTek Instruments) at l_exc_ = 610 nm and l_em_ = 680 nm. Standard curves were made with known added amounts of PS aggregation dissolved in the lysis buffer. The results were expressed as the number of PS aggregations per bacterium.

To understand the function of bacterial cell walls and membranes in the antibacterial process of PS aggregation, we measured the bacterial uptake of the PS aggregation using spheroplasts and intact bacteria as follows: (1) The PS aggregation was added to bacterial suspensions after the spheroplasts were prepared. (2) The intact bacteria bound to amounts of the PS aggregation. Aliquots of bacterial suspension (3.56 × 10^8^ CFU/mL) were incubated in 96-multi-well plates (Falcon) with the PS aggregation at different concentrations (0.2, 0.4, 0.8, 1.6, and 3.2 µM). The methods were the same as above.

### 4.15. Measurement of Dark Toxicity of PS Aggregation to Bacteria

The 0.05% tween20 or PBS suspensions of luminescent bacterial strains (*E. coli DH5α*) were placed in a 96-well white plate at a density of 10^6^ CFU/mL, and incubated in the dark for 5 min with ZnPc(Lys)_5_ at increasing concentrations (0.2 to 200 µM). The phototoxicity measurements of ZnPc(Lys)_5_ were performed under the same conditions with illumination (12.7 J/cm^2^). The luminescent intensity was recorded. The control without adding ZnPc(Lys)_5_ to the bacteria was performed in the dark and under light accordingly.

### 4.16. The Observation of Bacterial Growth Curve

Bacteria were allowed to grow into the mid-exponential growth phase in this experiment. The PS aggregation was diluted to a concentration of 0, 1/2-, 1-fold of its MIC value, respectively, using a culture medium with a final concentration of 1.2 × 10^6^ CFU/mL. Then bacteria were cultured under an anaerophilic environment at 37 °C and 150 rpm. The optical density at 600 nm was measured for each group every one hour, and each time a 100 μL bacterial suspension was taken into the 96-well plates for measurements until 12 h.

### 4.17. Outer-Membrane Disruption Assay

The outer-membrane permeabilizing ability was measured by fluorescent dye 8-Anilino-1-naphthalenesulfonic acid (ANS, Aladdin) [66] using *E. coli*. Bacterial cells from the mid-log phase were centrifuged and washed with a Tris buffer (10 mM Tris, 150 mM NaCl, pH7.2) and then re-suspended to an OD 600 of 0.20 in the Tris buffer. Different concentrations of the PS aggregation (0, 1/8×, 1/4×, 1/2 × MIC) were added to the bacterial suspensions and were incubated with bacteria for 3 min. Illumination (light dose = 0 and 12.7 J/cm^2^) was conducted before the probe ANS (0.565 mM) was added to the bacterial suspension. The extent of membrane disruption was observed as a function of PS concentration by the increase in fluorescence intensity at ~500 nm.

### 4.18. Bacterial Membrane Permeability

Membrane permeability of bacteria was determined by the N-phenyl-1-naphthylamine (NPN) uptake assay [67,68]. Bacterial cells in the mid-exponential growth phase were centrifuged, washed, and re-suspended with 5 mM HEPES (pH7.2) containing 5 μM carbonyl cyanide 3-chlorophenylyhdrazone (CCCP) to 0.5 OD at 600 nm. Various concentrations of the PS aggregation (0, 1/10-, 1/5-, 2/5-, 3/5-, 4/5-fold of MIC value of PS aggregation) were added to the bacterial suspension before illumination (light dose = 12.7 J/cm^2^) was conducted. NPN (10 µM) was added to the bacterial suspensions. The fluorescence intensity of the probe NPN was recorded. The excitation and emission wavelengths of NPN were set 350 and 420 nm, respectively.

### 4.19. RT-qPCR Analyses of Related Gene Expression

To quantify gene expression related to bacterial oxidative stress by real-time quantitative polymerase chain reaction (RT-qPCR), 1 mL of the bacterial culture in the late log phase of growth was treated with different amounts of the PS aggregation (final concentrations of 0, 1/8-, 1/4-, and 1/2-fold of its MIC value). A light dose of 15 J/cm^2^ was delivered three times with 5 J/cm^2^ for each illumination separated by 2 h. Total RNA was extracted immediately in the sixth hour using a TransZol RNA extraction kit (Invitrogen, Carlsbad, CA) according to the manufacturer’s instructions. The cDNA was synthesized from the RNA sample using a PrimeScript 1st Strand cDNA Synthesis Kit (Takara, Kyoto, Japan). qPCR was performed using the PowerUp SYBR Green Master Mix (Applied Biosystems) and Primers used in the RT-qPCR were listed (Appendix A). An Applied Biosystems QuantStudio 5 (Applied Biosystems) was used for qPCR. Data analysis was conducted using the 2^−∆∆CT^ method [36,69]. The relative expression ratio was represented as a log_2_ value in the histogram. A ratio greater than zero indicated up-regulation of gene expression, whereas a ratio less than zero indicated down-regulation. The gyrase gene gyrA [33,36,44], a housekeeping gene, was used as the reference for data normalization. The positive control using gene gyrA was conducted.

On the other hand, gene expression related to synthesis of the cell wall and membrane were measured using this method as above. Bacterial culture (1 mL) in the log phase of growth was treated with different light doses (0, 2, 4, and 6 J/cm^2^) at a 0.4 µM PS aggregation (pH7.2). Primers used in the RT-qPCR were listed (Appendix A). The gyrase gene gyrB [70], a housekeeping gene, was used as the reference for data normalization. The positive control using gene gyrB was conducted.

### 4.20. Hydrophobicity Measurement

The hydrophobicity of the bacterial surface was determined by bacterial adhesion to hydrocarbons (MATH) [65,71,72] with some modifications. Logarithmic phase bacteria were harvested, centrifuged, washed by a phosphate buffer solution (PBS, pH7.2), followed by 1 mM KNO_3_, and were re-suspended in KNO_3_ to 10^7^ CFU/mL for intact bacteria while the spheroplasts were washed and re-suspended in the spheroplast buffer. The suspension was mixed with various concentrations of the PS aggregation (0, 2, 4, and 8 µM). The hexadecane (0.2 mL) was added to the bacterial suspension (1.2 mL), vortexed for 10 min, and then settled for 30 min at 37 °C to see the phase separation. The aqueous phase at the bottom was removed, washed, and the absorbance at 600 nm was measured. At 600 nm, absorbance values of the PS aggregation bound to bacteria in the PBS were deducted after the absorbed PS aggregation was quantified using the method described above. All experiments were conducted in triplicate. The hydrophobic rate was calculated by the equation:Hydrophobic rate = 1 − OD_1_ / OD_0_ × 100%,
where OD_0_ is the absorption value at 600 nm without PS and OD_1_ is the absorption value at 600 nm with PS aggregation.

### 4.21. Zeta Potential Measurements

The zeta potential of the bacterial surface was determined by the Laser Particle Size and Zeta Potential Analyzer (BI-200SM, Brookhaven Instruments Corporation, New York, NY, USA) [73,74]. The logarithmic phase bacteria were harvested, centrifuged, and washed by the PBS and were followed by 1 mM KNO_3_ and re-suspended in KNO_3_ to 10^8^ CFU/mL for intact bacteria. The spheroplasts were washed and re-suspended in 1 mM KNO_3_ containing a spheroplast buffer. The bacterial suspension was mixed with different concentrations of the PS aggregation (0, 2, 4, 8, and 16 µM). The mixture was incubated for 3 min after the PS aggregation in the bacterial mixture was washed away. The zeta potential of the intact bacteria or spheroplasts was measured at room temperature. The zeta potential was determined from the mean of ten measurements.

### 4.22. Statistical Analysis

All experiments were conducted in triplicate at least. The data were expressed as mean ± standard deviation (SD). The statistical analysis was conducted by using Origin 8.5, Microsoft excel 2013, and SPSS 19.0. In all the statistical evaluations, *p* < 0.05 was considered statistically significant, and *p* < 0.01 was considered high statistically significant.

## 5. Conclusions

In this work, we reported a PS aggregate (Figure 8) with some unique properties: (1) The PS aggregate of photostability possessed a potent cellular effect at a micromolar concentration by PS monomers (Figure 8a). (2) The PS aggregate was disassembled upon contacting cells, generating PS monomers on the cellular surface (Figure 8b,c). (3) Interestingly, the disassembly of bound PS aggregates by cells was also facilitated by the presence of light (Figure 8d), resulting in the increase of the fractions of PS monomers on the cellular surface, the promotion of the PDT effect (Figure 8e), and the death of cells. (4) In PDT, the PS aggregate possessed no photobleaching, and it released PS monomers on the cellular surface, showing photodynamic activity leading to the destruction of cellular components.

## Figures and Tables

**Figure 1 molecules-28-04639-f001:**
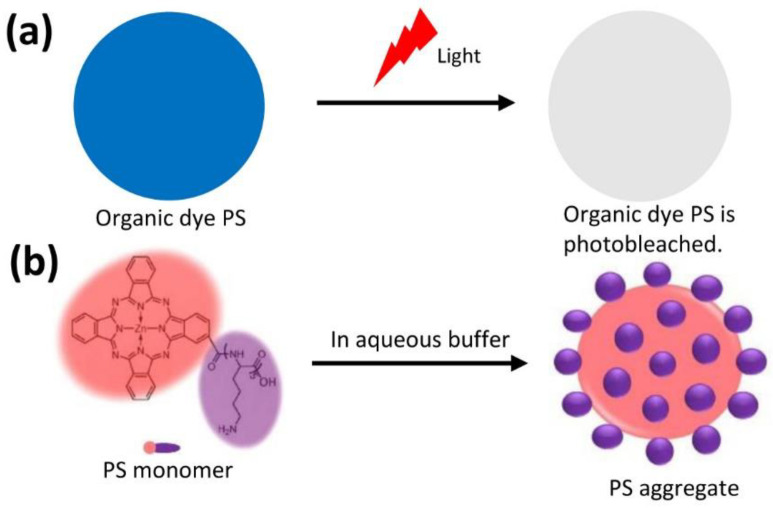
Photosensitizer (PS). (**a**) Organic dye PS was photobleached after illuminated. (**b**) PS monomer (ZnPc(Lys)_5_) was formed into PS aggregate in aqueous buffer.

**Figure 2 molecules-28-04639-f002:**
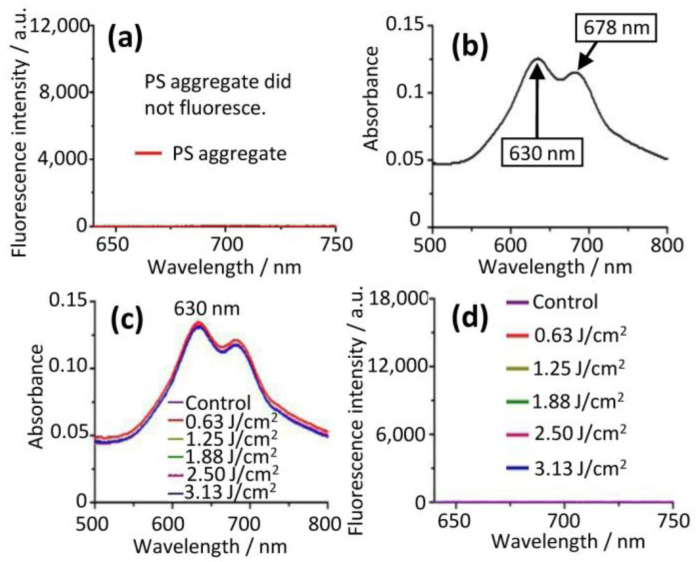
Photostability of PS aggregate in aqueous solution. (**a**) PS aggregate showed no fluorescence and (**b**) strong adsorption intensity at 630 nm. (**c**) UV/Vis absorbance spectra of PS aggregate did not change and (**d**) fluoresce with the increase of light doses (0 to 3.13 J/cm^2^). These reflected that PS aggregate possessed resistance to photobleaching.

**Figure 3 molecules-28-04639-f003:**
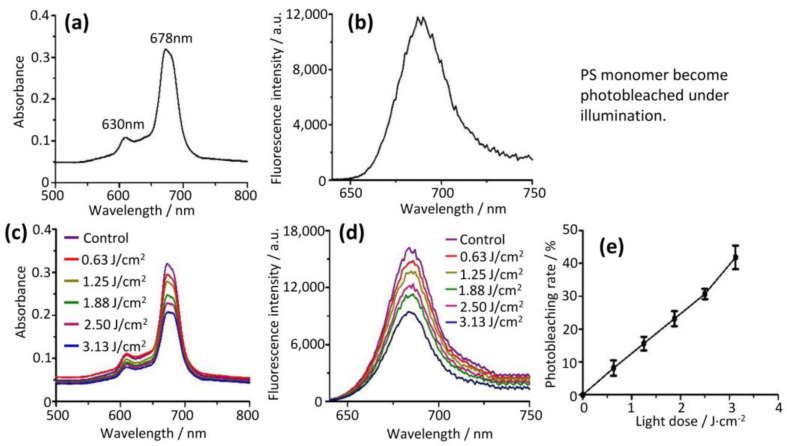
Photobleaching of PS monomer. (**a**) PS monomer had a Q band at 678 nm and (**b**) fluorescence spectra (l_exc_ = 610 nm, l_em_ = 680 nm). (**c**) Intensity of Q band at 678 nm and (**d**) fluorescence intensity significantly reduced with the increase of light doses (0 to 3.13 J/cm^2^), indicating that PS monomer possessed photobleaching. (**e**) Photobleaching rates of PS monomer increased upon increasing light doses.

**Figure 4 molecules-28-04639-f004:**
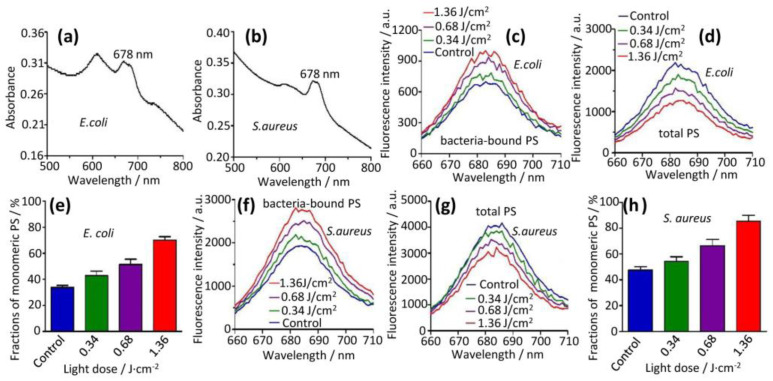
Cell-induced disassembly of PS aggregate and light-facilitated disassembly, generating PS monomer. Disassembly of PS aggregate was triggered in the presence of cell ((**a**) for *E. coli* and (**b**) for *S. aureus*). PS aggregate was incubated with bacteria ((**c**–**e**) for *E. coli* and (**f**–**h**) for *S. aureus*), illuminated with different light doses, and cell-bound PS were quantified by fluorescence (**c**,**f**). The mixtures of cell and bound PS were lysed in the lysis buffer and total PS was also quantified (**d**,**g**). PS monomer was generated from PS aggregate increased with light doses (**c**,**f**). However, the total PS amounts reduced upon light illumination (**d**,**g**), reflecting its partial degradation under light. The quantitation of the fractions of the cell-bound PS were shown in ((**e**) for *E. coli* and (**h**) for *S. aureus*).

**Figure 5 molecules-28-04639-f005:**
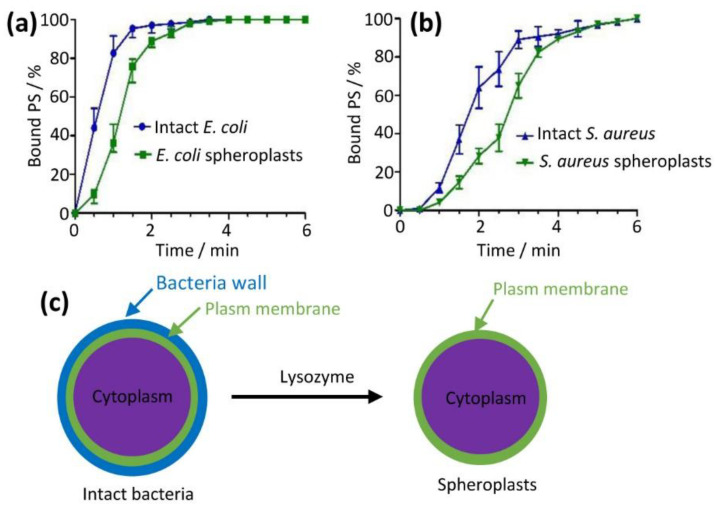
PS aggregate adsorption kinetics on cell ((**a**) for *E. coli* and (**b**) for *S. aureus*) measured by flow cytometry. Binding kinetics of PS aggregate on intact bacteria were faster compared to bacterial spheroplasts that were prepared as shown in (**c**).

**Figure 6 molecules-28-04639-f006:**
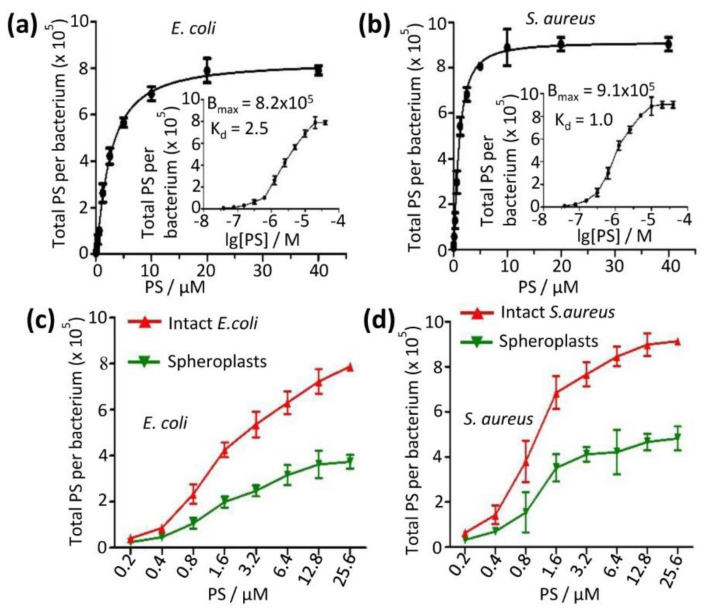
Full binding curves of PS to cell ((**a**) for *E. coli* and (**b**) for *S. aureus*) were measured to give estimates of the binding affinities (insets). Bound PS was evenly distributed between bacterial cell wall and bacterial membrane (total binding subtract by spheroplast) for both *E. coli* (**c**) and *S. aureus* (**d**) as shown by the binding curve. (**c**,**d**) The amount of bound PS to spheroplast membrane was about half of that to cell wall. PS at different concentrations was incubated with intact bacteria or spheroplasts, washed away, and dissolved in lysis buffer for quantization of total PS using fluorescence signals.

**Figure 7 molecules-28-04639-f007:**
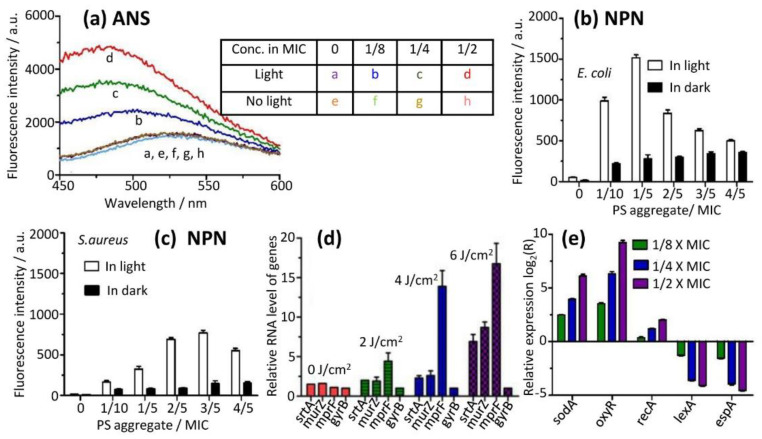
Disruption of cell membrane (**a**–**c**) and regulation of gene expression levels of related genes (**d**,**e**) during PDT of PS monomer released from its aggregate. (**a**) The fluorescence intensities of probe anilinonaphthalene-8-sulfonic acid (ANS) were measured with (trace a–d) or without light (e–h). (**b**,**c**) PS increased membrane permeability of *E. coli* (**b**) and *S. aureus* (**c**), measured under illumination using another fluorescence probe N-phenyl-1-naphthylamine (NPN). (**d**,**e**) Gene expression was quantified by RT-qPCR. Relative expression levels of genes related to cell wall and membrane biosynthesis were measured at different light doses (**d**). Expression levels of genes related to bacterial oxidative stress were measured under three sessions of separated illumination with various PS aggregate concentrations. The relative expression ratio was represented as a log_2_ value. A ratio greater than zero indicated up-regulation of gene expression, while a ratio below zero indicated down-regulation. In the experiments, MICs of PS aggregate towards *E. coli* and *S. aureus* were 1.6 and 0.8 µM, respectively.

**Figure 8 molecules-28-04639-f008:**
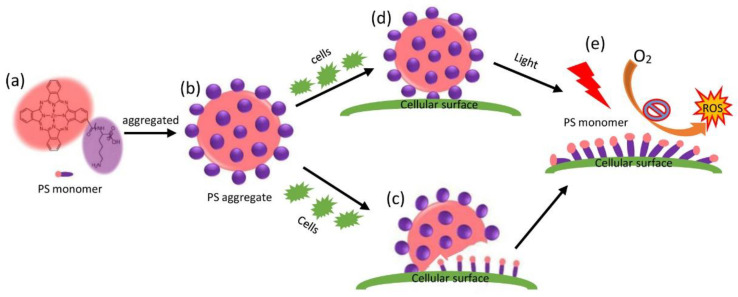
Cell or light-triggered disassembly of PS aggregation to generate PS monomer, leading to enhanced photodynamic inactivation toward cells. Monomeric PS (**a**) was formed into PS aggregation (**b**) to increase the photostability. (**c**) Cells induced disassembly of PS aggregation, generating monomeric PS on cellular surface. (**d**) Light also promoted the release of monomeric PS from bound PS aggregation. (**e**) Monomeric PS were bound to cellular surface, and were in charge of generation of ROS, leading to cellular death in PDT.

**Table 1 molecules-28-04639-t001:** Minimal inhibitory concentration (MIC, µM) and minimal bactericidal concentration (MBC, µM) of PS aggregate against *E. coli* and *S. aureus*.

Bacterial Strains	Incubation Time	
	MIC	MBC
*E. coli*	1.6	3.2
*S. aureus*	0.8	1.6

## Data Availability

Not applicable.

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
