# Peer review of "Monomer and Oligomer Transition of Zinc Phthalocyanine Is Key for Photobleaching in Photodynamic Therapy"

_molecules, 2023, doi:10.3390/molecules28124639_

Round 1

Reviewer 1 Report

This manuscript describes the monomer and oligomer transition of zinc phthalocyanine (ZnPc), which is important for photobleaching in photodynamic therapy of microorganisms. These demonstrated that ZnPc aggregate on bacterial surface photoinactivated bacteria via PS monomer during irradiation, where the photodynamic efficiency was retained without photobleaching. Mechanistic studies showed that ZnPc monomer disrupted bacterial membrane, and affected expression of genes related to cell wall synthesis, bacterial membrane integrity and oxidative stress.

Specific comments:

- Figures 2-4: Y axis: “absorbance/D.O.” should be changed to “absorbance”.

- 5.1. Light source. Fluence rate in mW/cm2 should be given.

- Line 99, Figure 3, etc. (ex=610nm, em=680nm) should be changed to (lexc=610nm, lem=680nm).

- Photobleaching rate should be also determined in microbial cell suspensions.

This manuscript describes the monomer and oligomer transition of zinc phthalocyanine (ZnPc), which is important for photobleaching in photodynamic therapy of microorganisms. These demonstrated that ZnPc aggregate on bacterial surface photoinactivated bacteria via PS monomer during irradiation, where the photodynamic efficiency was retained without photobleaching. Mechanistic studies showed that ZnPc monomer disrupted bacterial membrane, and affected expression of genes related to cell wall synthesis, bacterial membrane integrity and oxidative stress.

Specific comments:

- Figures 2-4: Y axis: “absorbance/D.O.” should be changed to “absorbance”.

- 5.1. Light source. Fluence rate in mW/cm2 should be given.

- Line 99, Figure 3, etc. (ex=610nm, em=680nm) should be changed to (lexc=610nm, lem=680nm).

- Photobleaching rate should be also determined in microbial cell suspensions.

Reviewer 2 Report

The manuscript submitted by Mingdong Huang et al. reports on the study of such important interrelated aspects of photodynamic action as the aggregation equilibria of Pc-based photosensitizers and their photobleaching in biological media. Aggregation is known to quench the formation of ROS, thus inhibiting their attack on both biological targets and PS itself. The authors have turned this disadvantage into an advantage by elegantly designing a dynamic Pc aggregate that can be safely delivered to the biological target even under light illumination, where it disaggregates and activates its photodynamic function, causing cell death. To support their hypothesis, the authors performed numerous mechanistic experiments, including spheroplasts tests, studies of cell membrane integrity during PDT action, etc. Biological targets causing cell death were identified by analysing the expression profiles of related genes.

I have a few minor questions that need to be addressed before this solid paper can be accepted for publication. 

Does dissolving the photosensitiser in pure Luria-Bertani medium without bacteria affect its aggregation state?

Have the authors attempted to characterise the aggregate size distributions using DLS or microscopy?

Figure 2 - The solvent used for photophysical characterisation should be stated in the caption.

Section 2.4 - I do not think the term "conformational state" is appropriate here. "Aggregation state" seems more appropriate, since the aggregation/disaggregation equilibrium is a driving force of the processes studied.

Minor manuscript formatting correction - there should be spaces between the text and the references in square brackets.

I think that some of the wording could be improved during the editing process, but overall the text is written in a clear and scholarly language.
